# Implicit Memory and Anesthesia: A Systematic Review and Meta-Analysis

**DOI:** 10.3390/life11080850

**Published:** 2021-08-19

**Authors:** Federico Linassi, David Peter Obert, Eleonora Maran, Paola Tellaroli, Matthias Kreuzer, Robert David Sanders, Michele Carron

**Affiliations:** 1Department of Anaesthesia and Intensive Care, Ca’ Foncello Treviso Regional Hospital, Piazzale Ospedale 1, 31100 Treviso, Italy; maraneleonora@gmail.com; 2Department of Anaesthesiology and Intensive Care, School of Medicine, Technical University of Munich, Ismaninger Str. 22, 81675 Muenchen, Germany; david.obert@tum.de (D.P.O.); m.kreuzer@tum.de (M.K.); 3Department of Developmental Psychology and Socialisation, University of Padova, Via Venezia 8, 35121 Padova, Italy; paola.tellaroli@gmail.com; 4Department of Anaesthetics, Royal Prince Alfred Hospital, Camperdown, NSW 2050, Australia; robert.sanders@sydney.edu.au; 5Department of Medicine, Anaesthesiology and Intensive Care, University of Padova, Via C. Battisti 267, 35121 Padova, Italy; michele.carron@unipd.it

**Keywords:** implicit memory, general anesthesia, awareness, anesthesia brain monitor, benzodiazepines

## Abstract

General anesthesia should induce unconsciousness and provide amnesia. Amnesia refers to the absence of explicit and implicit memories. Unlike explicit memory, implicit memory is not consciously recalled, and it can affect behavior/performance at a later time. The impact of general anesthesia in preventing implicit memory formation is not well-established. We performed a systematic review with meta-analysis of studies reporting implicit memory occurrence in adult patients after deep sedation (Observer’s Assessment of Alertness/Sedation of 0–1 with spontaneous breathing) or general anesthesia. We also evaluated the impact of different anesthetic/analgesic regimens and the time point of auditory task delivery on implicit memory formation. The meta-analysis included the estimation of odds ratios (ORs) and 95% confidence intervals (CIs). We included a total of 61 studies with 3906 patients and 119 different cohorts. For 43 cohorts (36.1%), implicit memory events were reported. The American Society of Anesthesiologists (ASA) physical status III–IV was associated with a higher likelihood of implicit memory formation (OR:3.48; 95%CI:1.18–10.25, *p* < 0.05) than ASA physical status I–II. Further, there was a lower likelihood of implicit memory formation for deep sedation cases, compared to general anesthesia (OR:0.10; 95%CI:0.01–0.76, *p* < 0.05) and for patients receiving premedication with benzodiazepines compared to not premedicated patients before general anesthesia (OR:0.35; 95%CI:0.13–0.93, *p* = 0.05).

## 1. Introduction

One of the most important goals of general anesthesia is to ensure the patient’s unconsciousness and unresponsiveness during induction and maintenance [1] as well as to achieve post-operative amnesia, usually detected by the absence of an explicit recall [2].

A popular assessment tool to evaluate responsiveness during anesthesia is the Observer’s Assessment of Alertness/Sedation Scale (OAA/S) [3]. The OAA/S score, which evaluates the patient’s behavioral response, speech, facial expressions, and ocular activity, ranges from a score of 5 (awake state and responsiveness) to 0 (unconscious state and unresponsiveness even to noxious stimuli). The use of neuromuscular blocking agents (NMBAs), decreasing muscle tone and preventing sudden muscle movements in response to a noxious stimulus, allows surgery under light general anesthesia, which can increase the risk of awareness and connected consciousness [1], even when an anesthesia brain monitor (ABM) is used [4]. While anesthesia awareness has an incidence of only 0.1–0.2% [1], connected consciousness, as detected by the isolated forearm technique, has a higher incidence, up to 34.8% [4]. The experience of such conscious episodes can lead to an increased risk of post-traumatic stress disorders [5,6].

The explanation for the low incidence of surgical recall may be due to the amnestic properties of the benzodiazepines and other hypnotics administered during general anesthesia [7]. Therefore, the gap between the incidence of anesthesia awareness and explicit recall, and the incidence of connected consciousness can be explained by the fact that patients may be aware of surgical events at the time they occur but may be unable to remember them later [8].

While explicit memory is recalled spontaneously, or may be provoked by postoperative events or questioning, implicit memory is not consciously recalled, and may affect behavior or performance at a later time. Implicit memory refers to any change in experience, thought, or action that is attributable to a past event [8] and can be detected with psychological tests, such as free association test, category member generation [9], word stem completion test [10], or the process dissociation procedure [11].

Several variables during anesthesia seem to interfere with implicit memory formation. They can be related to (i) anesthesia (like the type and dosage of hypnotic/analgesic drugs delivered for induction and/or maintenance, the usage of NMBA, anesthesia duration, or the use of ABM), (ii) the timing of the auditory task adopted for implicit memory formation (before skin incision or during surgical stimulation, repeated presentation of the words during the whole anesthesia period or otherwise), and (iii) the time point of implicit memory testing after the return of consciousness. However, the literature contains contrasting results regarding what has a significant impact on implicit memory formation during surgery and anesthesia.

Therefore, we conducted a systematic review and meta-analysis to evaluate the influence of these variables on implicit memory formation.

## 2. Materials and Methods

### 2.1. Search Strategy

We performed a systematic review and meta-analysis of previously published studies that investigated the occurrence of implicit memory after deep sedation or general anesthesia. Here, deep sedation refers to a spontaneously breathing patient with an OAA/S score of 0 or 1 (no response to mild prodding or shaking and no response to a noxious stimulus).

For the study design and the presentation of this report, we followed the Preferred Reporting Items for Systematic Reviews and Meta-Analyses (PRISMA, www.prisma-statement.org) (accessed on 16 August 2021) and we registered this meta-analysis in the PROSPERO international prospective register of systematic reviews (at the University of York, York, UK. Registration number: CRD42020170668).

We conducted a comprehensive English-language literature search of Medline, EMBASE and Google Scholar databases and used the following Medical Subject Headings (MeSH) terms: implicit memory, implicit recall, awareness, learning, free association, free recall, category member generation, word stem completion, and process dissociation procedure. These MeSH terms were also combined with the following additional terms using the ‘AND’ function: general anesthesia, anesthesia, sedation, OAA/S, intravenous, inhalational, minimum alveolar concentration (MAC). The search period included articles published between 1980 and March 2021. The date of the last search was 31 March 2021.

Two authors (FL, DO) independently identified potentially eligible articles by assessing titles and abstracts and reviewed full-text versions of these articles to select the studies to be included in this systematic review. In case of any disagreements regarding inclusion at either the title or abstract screening or the full-text review stage, a third author (MC) was consulted to resolve the issue.

### 2.2. Eligibility and Inclusion

We included controlled or observational trials if they involved adult patients (≥18 years) and listed implicit memory testing as a study endpoint.

A neuropsychologist (EM) critically evaluated the intraoperative methods, the post-operative psychological test batteries and the adopted criteria for the definition of implicit memory. Her evaluation was based on the analysis of the auditory task to be remembered that was presented to the patient during anesthesia, as well as on the psychological tests applied for the detection of implicit memory formation. Only if the authors performed the following standardized psychological tests for implicit memory detecting, the study was included: category generation task (CGT), sentence generation task (SGT), free association test (FAT), word stem completion test (WSC), word recognition test (WRT), general knowledge (GK), forced-choice recognition (FCR), preference, familiarity fame judgements (FFJ), process dissociation procedure (PDP) were considered [12,13,14].

We excluded studies that involved pediatric patients, brain and head-neck surgery, that did not clearly specify the used anesthetic regimen and the method for implicit memory testing (both, during intraoperative memory strategy (i.e., acquisition of words/sentences) and/or during the post-operative interview testing), or had no clearly defined criteria for implicit memory incidence. If studies evaluated implicit memory using (i) post-operative hypnosis, (ii) intra-operative behavioral or therapeutic suggestions, (iii) sounds (i.e., listening to a bird singing), or (iv) had not a control group, we excluded them. We also did not consider review articles and case reports.

### 2.3. Data Extraction

The following information was extracted from the articles and categorized: patient age (≤50 years or >50 years), American Society of Anesthesiologist physical status classification (only ASA I-II or also ASA III–IV), type of anesthetic approach (general anesthesia or deep sedation -OAA/S 0-1 with spontaneous breathing), duration of anesthetic procedure (≤60 min or >60 min), premedication with benzodiazepines (yes or no), anesthesia induction with benzodiazepines (yes or no), anesthesia maintenance (inhalational or intravenous), drugs (nitrous oxide use, benzodiazepines, opioids, NMBAs) used during anesthesia maintenance (yes or no), ABM use (yes or no), ABM-guided anesthesia (yes or no), surgical stimulation during the auditory task (yes or no), time point of auditory task delivery (throughout the entire anesthesia period or only for a time interval) (yes or no), time point of postoperative memory testing (≤24 h or >24 h).

We also estimated the anesthetic and analgesic level for each study. Therefore, two authors (FL and DO), independently categorized the anesthesia and analgesia regimen of each study (based on anesthetic and analgesic drugs and dosage applied during surgery) as ‘light’, ‘adequate’, or ‘deep’. Any disagreements were resolved by consensus with input from a third author (MC). Everyone was blinded to the outcome for this estimation. Therefore, another author (EM) prepared separate files not containing implicit memory rates, with a randomized sequence of anesthetic/analgesic regimens to be analyzed.

Every study considered for the meta-analysis was categorized according to these variables, and in case of different cohorts analyzed in the same trial, they were considered separately.

### 2.4. End-Points

The primary end-point was the incidence of implicit memory after general anesthesia or deep sedation. This incidence rate is defined as the rate between cohorts with patients developing an implicit memory to the total number of cohorts included in our analysis.

Secondary endpoints were the association between the considered variables and the implicit memory rates after general anesthesia and/or deep sedation or following only inhalational or only intravenous anesthesia maintenance regimens.

### 2.5. Assessment of Risk of Bias

The risk of bias of the included studies was assessed with the Cochrane risk of bias tool [15].

### 2.6. Statistical Analysis

For statistical analysis, we created a database for each endpoint (implicit memory after (i) general anesthesia and deep sedation, (ii) only general anesthesia, (iii) only inhalational maintenance anesthesia, or (iv) only intravenous maintenance anesthesia). The rates of implicit memory for these four endpoints were also evaluated according to anesthesia duration, benzodiazepine use (at premedication, at induction or during maintenance of anesthesia), the type of anesthetic regimen during maintenance of anesthesia (light or deep, with or without nitrous oxide), the type of analgesic regimen during maintenance of anesthesia (light or deep, with or without opioids), NMBA use, surgical stimulation during auditory task, the time point(s) of auditory task (during all the anesthetic period or only for a time interval), the usage of the ABM and ABM-guided anesthesia, and the timing of memory testing after surgery.

We estimated Odds ratios (ORs) and 95% confidence intervals (CIs) by normal approximation and we performed the meta-analyses of single proportions within a frequentist framework, using both random- and fixed-effect models. For continuity correction, we added 0.5 to the frequencies of every study, and we used a logit transformation to calculate the overall proportions. Confidence intervals for the individual studies were computed with the Clopper–Pearson method. The random-effect model was computed with inverse-variance weighting using the Der Simonian–Laird method to account for heterogeneity. We tested the heterogeneity across studies with Cochran’s Q statistic and the I2 statistic. A value of *p* < 0.1 indicated heterogeneity and we defined I2 > 50 to be substantial. All *p*-values were 2-tailed, with statistical significance set at <0.05. Funnel plots helped to visually assess potential publication bias, and formal linear regression tests of the funnel plot asymmetry were performed. We used R (version 3.3.1 for Windows) and the meta package for the analyses [16].

## 3. Results

From the 1167 potentially relevant studies that we initially identified in the literature, we excluded 1106 studies because they did not meet the inclusion criteria, were duplicates, or were incomplete in their methods or in their outcome data. Therefore, 61 studies, [9,10,11,12,17,18,19,20,21,22,23,24,25,26,27,28,29,30,31,32,33,34,35,36,37,38,39,40,41,42,43,44,45,46,47,48,49,50,51,52,53,54,55,56,57,58,59,60,61,62,63,64,65,66,67,68,69,70,71,72,73] involving a total of 3906 patients and 119 different cohorts, formed the basis for meta-analysis. The mean (±standard deviation) age and weight of the included patients was 45 (±10.49) years and 74.4 (±6.56) kg, respectively. Appendix A presents the general patient characteristics and the applied cohorts.

In general, the patients were scheduled mainly for general, gynecological, orthopedic, maxilla-facial, ophthalmic, plastic, vertebral, cardio-vascular, and urological surgery. A total of seven trials enrolled patients undergoing deep sedation without surgical stimulation.

Figure 1 presents the PRISMA flow diagram of the study selection process and Figure 2 shows the risk of BIAS summary of included studies. Among trials, we found a low risk of BIAS and a medium heterogeneity.

### 3.1. Implicit Memory Rates after General Anesthesia and OAA/S 1 Sedation

A total of 61 studies [9,10,11,12,17,18,19,20,21,22,23,24,25,26,27,28,29,30,31,32,33,34,35,36,37,38,39,40,41,42,43,44,45,46,47,48,49,50,51,52,53,54,55,56,57,58,59,60,61,62,63,64,65,66,67,68,69,70,71,72,73] (44 of them were randomized controlled trials (RCTs)) with a total of 3906 patients grouped into 119 cohorts evaluated implicit memory after deep sedation or general anesthesia. For 43 cohorts (36.13%), implicit memory formation was reported at postoperative evaluation.

We found a statistically significant difference in implicit memory formation for ASA physical status classification.

ASA III–IV patients had a significantly higher risk of implicit memory formation (OR [95%CI]: 3.48 [1.18–10.25], *p* < 0.05) compared to ASA I–II patients. Further, patients with deep sedation (OAA/S 0–1 and spontaneous breathing), compared to patients under general anesthesia, had a significantly lower risk of implicit memory formation (OR [95%CI]: 0.10 [0.01–0.76]) (Table 1).

### 3.2. Implicit Memory Rates after General Anesthesia

A total of 50 studies [9,10,11,12,17,18,19,20,22,23,24,25,26,28,29,30,31,32,33,34,35,37,38,39,40,41,42,43,45,48,49,50,51,52,53,54,55,56,58,60,61,62,63,65,68,69,70,71,73] (37 of them were RCTs) with a total of 3645 patients grouped into 103 cohorts evaluated implicit memory formation after general anesthesia. For 42 cohorts (40.78%) implicit memory formation at postoperative evaluation was reported.

Based on the variables analyzed, we did not find a statistically significant difference in implicit memory rates (Appendix A). Benzodiazepines premedication seems to have a protective role in preventing implicit memory formation, *p* = 0.05 (OR [95%CI]: 0.35 [0.13–0.92])

### 3.3. Implicit Memory Rates after General Anesthesia with Inhalational Maintenance

A total of 34 studies [9,10,11,12,17,18,20,21,22,23,24,25,26,28,29,30,31,33,35,38,41,42,43,45,49,52,58,60,63,65,69,70,71,73] (27 of them were RCTs) with a total of 2402 patients grouped into 67 cohorts evaluated implicit memory occurrence after general anesthesia with inhalational maintenance. Implicit memory formation at post-operative evaluation was found for 28 cohorts (41.79%).

There was no significant difference in the implicit memory rates for the analyzed variables (Appendix A). Benzodiazepines premedication seems to have a protective role in preventing implicit memory formation, *p* = 0.09 (OR [95%CI]: 0.26 [0.07–1.03])

### 3.4. Implicit Memory Rates after General Anesthesia with Intravenous Maintenance

A total of 22 studies [11,19,29,32,34,35,37,39,40,41,48,50,51,52,53,55,56,61,62,68,70,71] (15 of them were RCTs) with a total of 1243 patients grouped into 36 cohorts evaluated implicit memory occurrence after general anesthesia with intravenous anesthesia maintenance. Implicit memory formation at post-operative evaluation was described for 14 cohorts (38.89%).

There was no significant difference in the implicit memory rates for the analyzed variables detection (Appendix A).

## 4. Discussion

This systematic review and meta-analysis suggests that while the patients’ comorbidity may increase the likelihood of implicit memory formation, general anesthesia with premedication may decrease the likelihood of implicit memory formation. Further, deep sedation (Observer’s Assessment of Alertness/Sedation of 0–1 with spontaneous breathing) seems more protective in implicit memory formation than general anesthesia regimens.

### 4.1. Patient Characteristic Considerations

According to this meta-analysis, an ASA physical status III–IV, compared to ASA physical status I–II, significantly increased the risk of implicit memory formation (OR 3.48). For several reasons, this may mainly be attributed to the tendency to use light anesthesia in these patients. The patients’ comorbidity (e.g., presence of limited cardiac reserve) and other factors (e.g., reduced cardiac output, hypovolemia, hemodynamic instability) may influence anesthetic delivery and predispose the patient to a period of light anesthesia [74]. Light anesthesia, the increased anesthetic requirement of some patients (e.g., younger age, tobacco smoking, long-term use of certain drugs (alcohol, opiates, or amphetamines), and machine malfunction or misuse, resulting in an inadequate delivery of the anesthetic, are usually associated with the risk of awareness as well as explicit and implicit memory formation [75].

Even if the impact of physiological changes on the pharmacokinetics/pharmacodynamics of anesthetic drugs as well as on ABM observed with aging [76,77,78] was postulated to influence explicit and implicit memory formation [79], in our study, aging seems not to be a risk factor for implicit memory occurrence. The age-induced decline of implicit memory (e.g., priming) is less than the decline of explicit memory (e.g., recognition) [79]. Explicit memory and implicit memory involve different parts of the brain: while the hippocampus is necessary for explicit memory, the amygdala, a small structure located near the hippocampus, is essential for implicit memory formation [80]. A surgery-induced stress response causing elevated norepinephrine levels and amygdala activity (in addition to other regions) would facilitate the implicit learning of emotionally negative information presented during anesthesia [49]. Although the amygdala remains structurally preserved during normal aging, functional magnetic resonance imaging studies revealed an age-related decrease in its activating response to negative stimuli [81]. Hence, this dichotomous age-induced influence on the amygdala could justify the different literature findings on the impact of aging on implicit memory after anesthesia.

### 4.2. Anesthesiologic Considerations

Implicit memory formation was higher among the general anesthesia regimens (40.78%) than among the deep sedation regimens (6.25%) evaluated. A high percentage of general anesthesia cohorts used NMBAs (94.1%) and opioids (87.3%), whereas the deep sedation cohorts were conducted without NMBAs (100%) and, with exception of two studies [46,57], without opioids (87.5%). These findings support the literature and reinforce the potential role of general anesthesia in implicit memory occurrence [68]. Sedation is a continuum that proceeds from minimal to deep levels in a dose-response manner, which may ensure the immobility of the patient following repeated or painful stimulation. Instead, general anesthesia is distinct in which unconsciousness, analgesia, and immobilization are provided by different drugs [82]. Both strategies may, then, ensure immobility during the procedure, but with different levels of suppression of consciousness and memory formation [83]. Administering NMBAs for the immobility of the patient could favor light general anesthesia, which, in turn, increases the incidence of explicit recall and awareness [75,84,85]. In our meta-analysis, the use of NMBAs was shown to have the potential of increasing the likelihood of implicit memory (OR 3.02).

In our study opioids were not significantly related to implicit memory formation, supporting literature findings that suggest that implicit memory formation should be considered despite the use of opioids [68,85,86]. Recent evidence shows that hippocampal µ-opioid receptors on GABAergic neurons mediate the stress-induced impairment of memory retrieval [87]. However, opioids, which reduce the amount of anesthetic drugs necessary for the loss of consciousness [88], may also increase the risk of light anesthesia [1], Furthermore, an anesthetic regimen with a low opioid dose, perhaps with less potency such as fentanyl or sufentanil compared with remifentanil, would not prevent endogenous catecholamine release in response to surgical stimulation, which would, in turn, enhance implicit memorization [68].

On the other hand, most of the deep sedation regimens were based on propofol or midazolam, two γ-aminobutyric acid type A receptor agonists with an amnestic effect [80], which were administered in the absence of opioids and NMBAs, at a dosage to prevent a reaction following repeated or painful stimulation.

Even if a difference in implicit memory rates between light and deep general anesthesia regimens was not found, it may depend on the heterogeneity of the studies and the method to establish a ‘light’ or ‘deep’ anesthetic regimen. A ‘subjective’ evaluation of an anesthetic plan, based on the type and the dosage of drugs used during the surgery, was adopted by three authors. An anesthetic regimen judged ‘adequate, not light’, might be sufficient to prevent conceptual priming but insufficient to prevent perceptual priming, which represents the basis of implicit memory generation, during surgery [89]. High noxious stimulation, such as surgical stimulation, may cause hypnotic state fluctuations associated with very short periods of awareness and the reinforcement of memory formation caused by amygdala stimulation via endogenous stress hormone release, which finally leads to implicit memory formation [49,68]. This may also explain why ABM (and ABM-guided anesthesia) has no effect in preventing implicit memory formation. The close proximity of the sensor to the rostral structures of the brain allows detection of EEG signals correlated with the neural functions of the cerebral cortex, which relate mainly to wakefulness and awareness [90]. However, some brain areas (e.g., cerebellum, striatum, hippocampus, amygdala, substantia nigra, ventral tegmental area of the midbrain), that are essential for the formation, re-organization, consolidation, and storage of memory and involved in implicit memory formation, are excluded by brain monitoring [91,92].

Premedication with benzodiazepines, well known for their amnestic properties and their effect on memory formation by impairing the ability to acquire new information [93], can help to avoid awareness and it is particularly suggested when light anesthesia is anticipated [90], Our meta-analysis showed that premedication with benzodiazepines may also avoid implicit memory formation during general anesthesia (OR = 0.35). The potential benefit seems greater during the maintenance of general anesthesia with inhalational agents (OR = 0.26) rather than with an intravenous agent (OR = 0.48). Even if we did not find statistically differences in implicit memory formation comparing inhalational versus intravenous general anesthesia alone, this may result from the different impacts of inhalational and intravenous drugs on memory and a synergism between benzodiazepines and inhalational anesthetic drugs in preventing implicit memory [93,94], It has been shown that sub-hypnotic doses of propofol block hippocampal but not amygdala response to emotionally arousing memory tasks [95], Instead, inhalational anesthetic agents, such as halothane, isoflurane, sevoflurane, and desflurane may impair learning and memory at sub-hypnotic concentrations [94] by slowing the hippocampal θ-rhythm [96] that is implicated in memory processing through activation of plasticity and interregional signal integration, which impairs hippocampus-dependent implicit memory and learning formation [97], This may explain why, in our meta-analysis, premedication with benzodiazepines resulted in greater benefits in inhalational anesthesia than in intravenous anesthesia.

### 4.3. Auditory Task Characteristic Considerations

The considerations made so far help form an understanding of the potential role of the time point the auditory task was delivered (continuously or not continuously during anesthesia, during or not during surgical stimulation) for implicit memory formation. While the auditory task was presented during surgical stimulation in only 25% of the deep sedation regimens, it was presented in 88.35% of the total general anesthesia regimens. In addition, no tasks were executed continuously during deep sedation regimens, whereas 23.71% of those during general anesthesia were. Based on our data, implicit memory is not induced by repetitive auditory stimulation during surgery and requires top-down processing, which is suppressed by the respective anesthetic plan [98]. However, while the timing of the presentation of the auditory task is not significantly related to implicit memory formation during general anesthesia, its role cannot be completely overlooked [46,48]. Inadequate analgesia regimens, particularly during surgical stimulation, can lead to fluctuations of consciousness that, while not sufficient for explicit memory formation, can be responsible for implicit memory [52], which is enhanced by the release of endogenous catecholamine and the subsequent simulation of the amygdala [56]. In such situations, auditory stimulation (e.g., listening to auditory tasks) may predispose a patient to implicit memory formation even while unconscious [5,82], Even if not statistically different, listening to the auditory task during surgical stimulation has an OR of 2.53 in implicit memory formation, and this, along with the different surgery stimulus allowed in general anesthesia in respect to deep sedation (with a lower endogenous catecholamine release during deep sedation surgery reasonably possible), might have played a role in our results reflecting higher cases of implicit memory during general anesthesia than deep sedation.

### 4.4. Limitations

This study has some limitations. First, there were not enough deep sedation regimens to perform a subgroup analysis such as for only general anesthesia, so we were unable to analyze each variable compared to the implicit memory rate for only deep sedation regimens. Second, although a neuropsychologist reduced heterogeneity by excluding some implicit memory detection techniques (e.g., hypnosis, behavioral and therapeutic suggestions, listening to nature sounds as hearing stimuli), not all of the studies considered adopted the same standardized tests at the same time intervals (however, this variable was considered and found not to be significant). Additionally, a high degree of heterogeneity among different forms of conduction of general anesthesia was found, especially with respect to the types and doses of drugs used, although this diversity could reflect daily anesthetic practices.

## 5. Conclusions

In our meta-analysis, patients with a higher ASA physical status (III–IV) were at a significantly higher likelihood of implicit memory formation. Deep sedation was associated with a significantly lower incidence of implicit memory formation. Meanwhile, the type of anesthetic/analgesic regimen for maintenance does not significantly impact on implicit memory formation. Premedication with benzodiazepines before inhalational general anesthesia seems to be more protective than intravenous one.

## Figures and Tables

**Figure 1 life-11-00850-f001:**
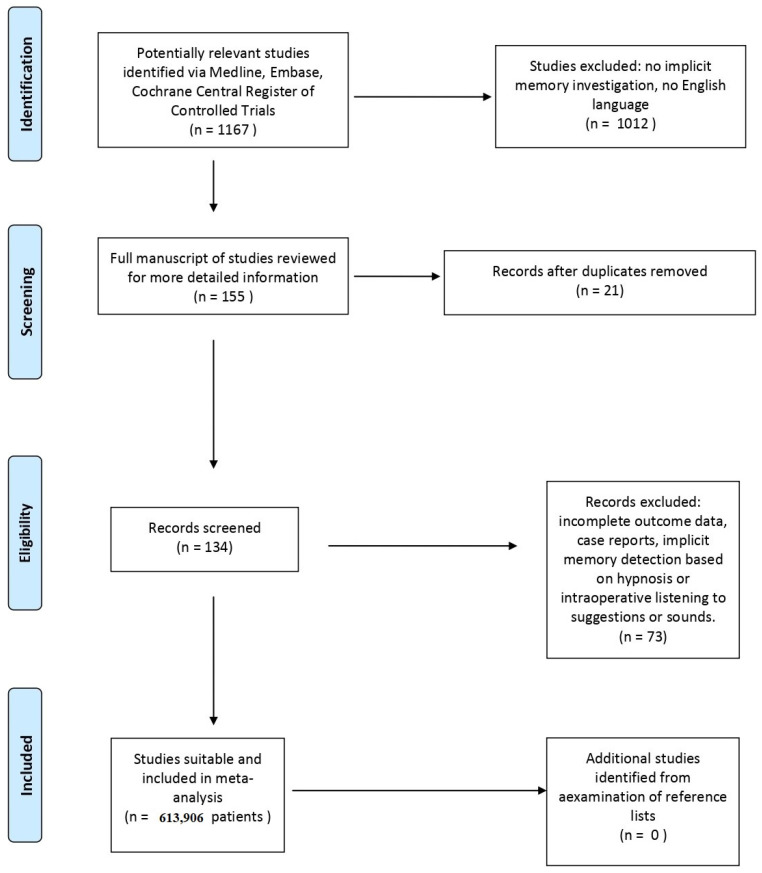
PRISMA flow diagram of the considered trials.

**Figure 2 life-11-00850-f002:**
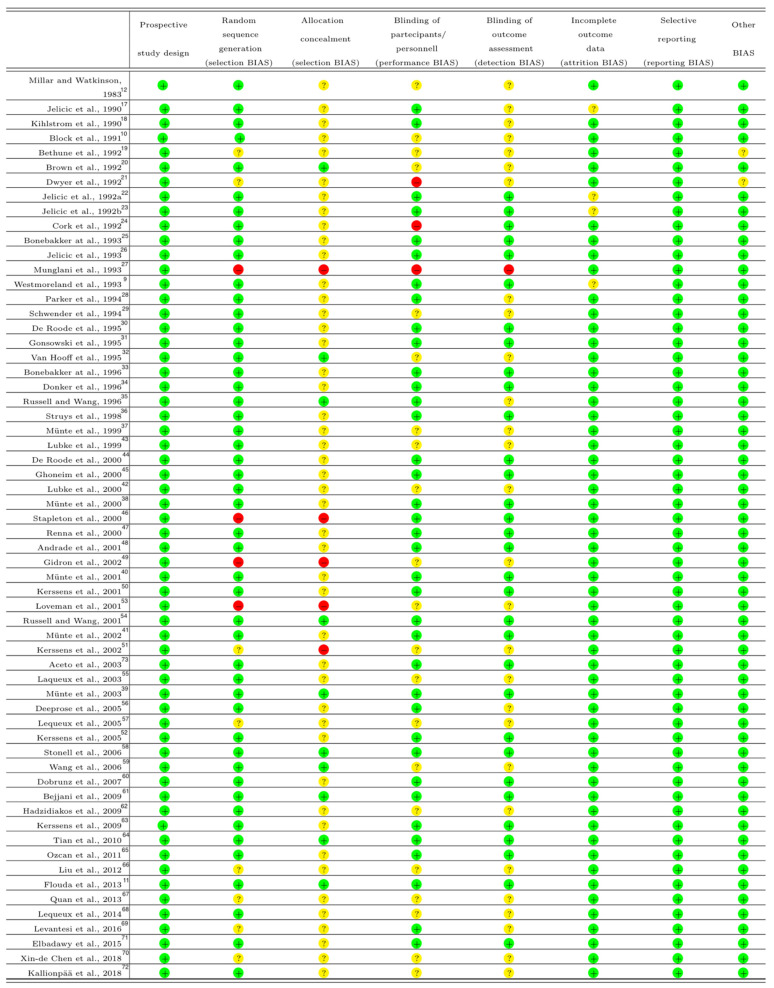
Risk of bias summary of included studies. Green circle for low risk of BIAS; red circle for high risk of BIAS; yellow circle for unclear risk of BIAS.

**Table 1 life-11-00850-t001:** Distribution of cohorts among the considered variables for general anesthesia and deep sedation (OAA/S 0–1) regimens.

Variable	Implicit Memory					
	Yes	No	OR	L95%CI	U95%CI	*p*-Value
**Age**						
≤50 years	31	58				
>50 years	11	10	2.06	0.79	5.38	0.22
**ASA**						
I–II	23	56				
III–IV	10	7	3.48	1.18	10.25	0.04
**Type of anesthesia**						
General anesthesia	42	61				
Deep sedation	1	15	0.10	0.01	0.76	0.017
**Duration**						
≤60 min	4	16				
>60 min	25	39	2.56	0.77	8.56	0.20
**Premedication**						
No benzodiazepines	34	52				
Benzodiazepines	7	23	0.47	0.18	1.20	0.16
**Induction**						
No benzodiazepines	40	70				
Benzodizepines	3	6	0.88	0.14	4.37	1.0
**Maintenance**						
Intravenous	29	42				
Inhalational	14	34	1.68	0.77	3.67	0.26
No N_2_O during maintenance	21	46				
N_2_O during maintenance	22	30	1.61	0.76	3.42	0.30
No benzodiazepines	42	74				
Benzodiazepines	1	2	0.88	0.02	17.42	1.0
No opioids	6	23				
Opioids	37	53	2.68	0.99	7.21	0.08
No NMBA	4	18				
NMBA	39	58	3.02	0.95	9.62	0.09
No light anesthetic regimen	29	52				
Light anesthetic regimen	14	24	1.05	0.47	2.33	1.0
No deep anesthetic regimen	14	25				
Deep anesthetic regimen	29	51	1.02	0.46	2.25	1.0
No light analgesic regimen	25	41				
Light analgesic regimen	18	35	0.84	0.40	1.80	0.80
No deep analgesic regimen	18	25				
Deep analgesic regimen	25	51	1.02	0.46	2.25	1.0
**Monitoring**						
No ABM monitoring	24	48				
ABM monitoring	15	28	1.07	0.48	2.38	1.0
No AMB-guided anesthesia	36	57				
ABM-guided anesthesia	7	18	0.62	0.23	1.62	0.45
**Listening to the auditory task**						
No during surgical stimulation	5	19				
During surgical stimulation	38	57	2.53	0.87	7.37	0.13
No during all the maintenance period	31	52				
During all the maintenance period	7	16	0.73	0.27	1.98	0.71
**Timing of memory testing**						
≤24 h	31	54				
>24 h	12	20	1.05	0.45	2.42	1.0

ASA: American Society of Anesthesiologist physical status classification; deep sedation: Observer Assessment of Alertness/Sedation score 0–1; NMBA: neuromuscular blocking agent; ABM: anesthesia brain monitor; OR: odds ratio; L95%CI and U95%CI: lower limit and upper limit of the 95% confidence interval (CI).

## Data Availability

The data presented in this study are available in Appendix A.

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
