# Peer review of "Implicit Memory and Anesthesia: A Systematic Review and Meta-Analysis"

_life, 2021, doi:10.3390/life11080850_

Round 1

Reviewer 1 Report

I read the manuscript “Implicit Memory and Anaesthesia: a systematic review and meta-analysis” with much interest. The authors identified a gap in the literature and were able to sum the confusing literature in an effective and understandable way. The literature search and inclusion criteria are appropriate, the data base is significant and the conclusions are clear and useful. The manuscript is well written and the data fairly presented. There are a few issues that need to be corrected in the presentation and analysis of the data, but overall, this work is definitely worthy of publication.

Below is a detailed description of the issues that I would like the authors to consider prior to publishing the manuscript.

Major comments:

- As much as I like the idea, there are no 119 different anesthetic regimens, and claiming that the study compared that many regimens seems misleading. Furthermore, there are not enough patients in the cohort to test that many regimens (or variables). I could accept 2-5 anesthetic regimens (which can be divided into sedation / GA, volatile/TIVA and +/- NMBA). However, all other (minor) changes in the anesthetic drug combinations and techniques do not create different anesthetic regimens, but are merely measured variables of the administered anesthesia. Claims such as “implicit memory occurred in 43 anesthetic regimens” is unhelpful – detailing the protective effect of benzodiazepines or the effects of specific drugs (or drug classes) on implicit memory formation is much more accurate and helpful. I would suggest avoiding the “regimens count” and concentrating on which common features (drugs, techniques) these regimens have.

- Table 1 is unclear – what are the numbers under the yes\no columns? Percentages? Number of cases (out of how many)? As far as I can tell the numbers don’t add up to anything. Please indicate what the numbers represent (e.g. number of patients with a specific feature that developed implicit memory). If these are patient counts – please add the denominator, if these are percentages – please add the actual event number. In any case – it should be clear from the text and legend what the numbers represent.

- Figure 1 – The figure is unclear – if the search result discovered 1167 studies, how could 2181 be excluded? (and leave 1012 studies)? If out of 1012 studies 21 were removed for duplications, how come we’re left with only 134 studies? Where did all the other studies go?

- Tables 2-3-4 seem to be redundant with no significant information added (and very few variables with borderline significance). I wound suggest detailing the interesting results in the text and moving these tables to the supplementary material.

- Usually, the differentiation between GA and sedation is by whether an AW manipulation was performed. Often times we see patients under GA, spontaneously breathing with no AW manipulation that are erroneously classified as sedation, while other times patients under much lighter sedation are given muscle relaxants and intubated and thus considered under GA. If you consider this issue, it makes sense that AW manipulation (a very strong stimulus) is a risk factor for implicit memory formation. This makes the explanation of why sedation is a protective factor easy. I would suggest considering that AW manipulation is just another variable of the anesthesia, rather than dividing patients into sedation vs. GA.

- I am not sure that the authors have the granular data to do the calculations, but the presented data seems to call for multivariate analysis that will help identify the truly significant variables. Is it possible to obtain the data to perform this analysis?

- The authors suggest that sicker patients are at risk due to shallow anesthesia. In this case – ABM should have a protective effect in this group. If so, was this reported in any of the screened studies?

Minor comments:

Page 2 lines 96-97 – there’s an error in the search period specification (unless 198312 is a date), please correct.

Page 3 line 117 – need to add “studies” at the beginning

Page 6 line 177 – I believe that it should be 12.5 (not 12>50)

Why is Figure 1 following figure 2 (and not presented before?)

Reviewer 2 Report

Manuscript ID: life-1330508 

Title: "Implicit memory and anaesthesia: a systematic review and meta-analysis"

The aim of this research was to evaluate the effect of anaesthesia on implicit memory formation. To this aim, 61 studies were selected by databases. Overall, results indicate that general anaesthesia with premedication may decrease the likelihood of implicit memory formation. Moreover, deep sedation seems to be more protective in implicit memory formation than general anaesthesia.

The topic of the present work is interesting. The manuscript is clearly presented and methodology sounds adequate.

I only have few suggestions.

The rationale for some choices is lacking. For example, why did age is categorized in <50 and >50? The same question about the time (<60 minutes or > 60 minutes).

What does the table at page 5 represent? The legend is lacking.

Data reported in Table 1 are non-clear. For example, the crossing age with yes/no shows 31-58-11-10. The sum is 110. What does such number mean? Studies? But the studies selected are 61. I feel that Authors should explain it better.

Considering that multiple comparisons are performed on the same sample, I ask myself whether it should be better to adopt the Bonferroni correction in order to decide the p-value.
